# REAL&SYNTHETIC DATASET AND THE LINEAR ATTENTION IN IMAGE RESTORATION

## ABSTRACT

Image restoration (IR), which aims to recover high-quality images from degraded inputs, is a crucial task in modern image processing. Recent advancements in deep learning, particularly with Convolutional Neural Networks (CNNs) and Transformers, have significantly improved image restoration performance. However, existing methods lack a unified training benchmark that specifies the training iterations and configurations. Additionally, we construct an image complexity evaluation metric using the gray-level co-occurrence matrix (GLCM) and find that there exists a bias between the image complexity distributions of commonly used IR training and testing datasets, leading to suboptimal restoration results. Therefore, we construct **a new large-scale IR dataset called ReSyn**, that utilizes a novel image filtering method based on image complexity to achieve a balanced image complexity distribution, and contains both real and AIGC synthetic images. From the perspective of measuring the model's convergence ability and restoration capability, we construct **a unified training standard** that specifies the training iterations and configurations for image restoration models. Furthermore, we explore how to enhance the performance of transformer-based image restoration models based on linear attention mechanism. We propose **RWKV-IR**, a novel image restoration model that incorporates the linear complexity RWKV into the transformer-based image restoration structure, and enables both global and local receptive fields. Instead of directly integrating the Vision-RWKV into the transformer architecture, we replace the original Q-Shift in RWKV with a novel Depth-wise Convolution shift, which effectively models the local dependencies, and is further combined with Bi-directional attention to achieve both global and local aware linear attention. Moreover, we propose a Cross-Bi-WKV module that combines two Bi-WKV modules with different scanning orders to achieve a balanced attention for horizontal and vertical directions. Extensive experiments demonstrate the effectiveness and competitive performance of our RWKV-IR model.

## 1 INTRODUCTION

Image restoration (IR), which aims to recover high-quality images from low-quality degraded inputs, is a crucial task in modern image processing. This field encompasses various sub-tasks, including super-resolution, image denoising, and compression artifacts reduction. Recently, the advancements of deep learning techniques, such as Convolutional Neural Networks (CNNs) Dai et al. (2019); Dong et al. (2014); Lim et al. (2017); Zhang et al. (2017a; 2018b) and Transformers Chen et al. (2021; 2023a;d); Li et al. (2023a); Liang et al. (2021), have significantly enhanced image restoration performance, driving continuous progress in this field. Reviewing the previous IR methods, more complex and deeper models Zhou et al. (2023); Chen et al. (2023a) often achieve better performance.

To meet the data requirements for IR model training, a large number of images need to be collected to construct a paired training dataset. Due to limited photography and compression techniques, the images in the previous training datasets often have the problem of blurring or noise, meanwhile the images in some test datasets have more image details, this causes the domain gap (different image complexities) between the commonly used training and test datasets Li et al. (2023b). And most datasets Timofte et al. (2017); Li et al. (2023b) focused on collecting a large number of images of high resolution, with few datasets considering how to measure and address this domain gap.

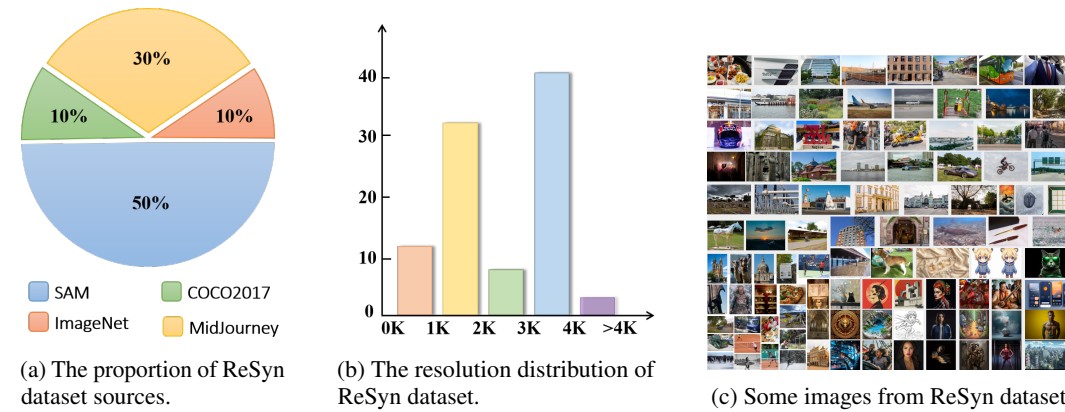

(a) The proportion of ReSyn dataset sources.

(b) The resolution distribution of ReSyn dataset.

(c) Some images from ReSyn dataset.

Figure 1: The diversity analysis of our ReSyn dataset. It contains both real and synthetic images from a variety of data sources and covers a wide range of resolutions.

In this paper, we construct an image complexity metric based on the Gray-Level Co-occurrence Matrix (GLCM) analysis method, and reveal a significant bias in image complexity distribution within classic IR training datasets and test datasets. We further utilize this metric as a criterion for filtering the image restoration dataset and construct a new image restoration dataset, which is called **ReSyn dataset**. In many previous datasets Li et al. (2023b); Timofte et al. (2017), resolution and Bits Per Pixel (BPP) have been used as important criteria for image filtering, but they are insufficient for constructing *image complexity balanced datasets*. To this end, we further utilize the GLCM-based image complexity metric we proposed to filter and retain some images of medium resolution but with high image complexity. Moreover, with the rapid development of AI-generated content (AIGC), there is a surging demand for synthetic image restoration. We consider the generated images as an essential part of the dataset and filter them in the same manner, which also enriches the sources of our dataset. The final ReSyn dataset comprises 12,000 images, with 30% being high-quality synthetic images sourced from the web. Our ReSyn dataset also presents a wide range of image resolutions, ranging from 0.25K to 4K, and originates from various sources. Experiments have also demonstrated the effectiveness of this dataset construction approach.

We also review the training processes of previous image restoration models and find that there is a lack of a **unified IR training benchmark**, *i.e.,* the training iterations and configurations are not unified. Considering the model's convergence and restoration capability, we construct a set of training standards. To measure the model's convergence capability, we use a shorter number of training iterations; to measure the restoration capability, we use a longer number of training iterations. This combination of short and long training iterations allows users to have a more comprehensive understanding of the model's capabilities, facilitating the selection of the model. We conduct a comprehensive evaluation of state-of-the-art IR models using the unified training standard, on our ReSyn dataset and other commonly used IR datasets. Both the ReSyn dataset and the constructed unified training standard form our proposed benchmark.

When comparing various image restoration models, we notice that the linear attention mechanism (*e.g.,* Mamba-based IR models) has a great potential for enhancing the effective receptive fields of models. Therefore, we aim to incoperate a linear complexity attention mechanism, RWKV Peng et al. (2023), with the image restoration models. We propose **RWKV-IR**, a novel image restoration model with both global and receptive fields, which can effectively restore low-quality images with linear computational complexity. Our RWKV-IR consists of three stages: shallow feature extraction, deep-feature enhancing, and HQ image reconstruction modules; and we incorporate RWKV into the deep-feature enhancing. We first introduce the Vision RWKV module to extract the deep image features, where a Spatial Mix Layer is employed to enable our model with global receptive fields with only linear complexity. Then, to enhance the modeling of relationships in local receptive fields and eliminate the negative effects caused by the original Q-shift operation, we exploit the characteristics of the local receptive fields in convolutional operations, and propose a **Depth-wise Convolution Shift (DC-shift)** module, as a replacement of the Q-shift in the original RWKV. Moreover, the Bi-WKV method of original Vision-RWKV has an unbalanced position embedding, paying more attention to the horizontal direction and less attention to the vertical direction, which is not suitable

for the IR tasks. We propose a **Cross-Bi-WKV module**, which combines two Bi-WKV modules with different scanning orders to achieve a balanced attention to surrounding features. By cross-scanning and synchronizing the calculations of the two Bi-WKV modules in both horizontal and vertical directions, we achieve a balanced attention to all four directions. Extensive experiments demonstrate the effectiveness of our RWKV-IR.

The main contributions of this paper are summarized as follows:

- We propose a comprehensive benchmark for image restoration tasks, which includes a novel large-scale benchmark dataset, and a unified training standard that specifies the number of training iterations and the configuration of batch size. We conduct a comprehensive evaluation of state-of-the-art IR models using the unified training standard on the new dataset and other commonly used IR datasets.

- We construct a large-scale dataset called ReSyn, that integrates both real and synthetic images. This dataset encompasses a variety of data sources and utilizes novel image filter methods based on our newly proposed GLCM-based image complexity metric.

- We design RWKV-IR, a novel image restoration model with both global and local receptive fields, which can effectively restore low-quality images with linear computational complexity. We replace the the original token-shift method (Q-shift) with Depth-wise Convolution shift for local dependencies modeling, and proposes Cross-Bi-WKV to replace Bi-WKV for a more balanced attention for horizontal and vertical directions, which enables RWKV to be effectively transferred to IR models.

## 2 RELATED WORKS

### 2.1 IMAGE RESTORATION

Image restoration has witnessed significant progress with the advent of computer vision Liang et al. (2021); Zhang et al. (2022a); Chen et al. (2023b); Guo et al. (2024), exemplified by pioneering CNN-based methods like SRCNN Dong et al. (2014), DnCNN Zhang et al. (2017a), and ARCNN Dong et al. (2015), targeting super-resolution, denoising, and artifact reduction Kim et al. (2016); Zhang et al. (2021c); Cavigelli et al. (2017); Wang et al. (2018); Zhang et al. (2018d); Lai et al. (2017); Wei et al. (2021); Fu et al. (2019); Zhang et al. (2018c); Dai et al. (2019). Despite their success, CNN-based approaches often struggle to model global dependencies effectively. Meanwhile, Transformers, proven competitors to CNNs in various computer vision tasks Carion et al. (2020); Dosovitskiy et al. (2020); Liu et al. (2021); Zhang et al. (2021a; 2023b), show promise in restoration tasks. However, they encounter challenges due to the quadratic computational complexity of the attention mechanism Vaswani et al. (2017). Strategies like IPT Chen et al. (2021) and SwinIR Liang et al. (2021) address this by employing patch-based processing and shifted window attention. But the trade-off persists between efficient computation and global modeling Zhang et al. (2023a); Chen et al. (2023a); Li et al. (2021); Chen et al. (2023d); Zamir et al. (2022); Chen et al. (2023c). Recently, MambaIR Guo et al. (2024) has been proposed to incorporate Mamba Gu & Dao (2023); Liu et al. (2024) into the image restoration task, which globally processes the image features with only linear computational complexity. Following this trend, this paper explores the possibility of integrating another linear attention mechanism, RWKV (which has shown better performance than mamba in other vision tasks Fei et al. (2024); He et al. (2024); Gu et al. (2024), into image restoration models.

**Single Image Restoration Datasets.** Learning-based image restoration methods rely on the external training dataset to learn the mapping between degraded and GT images. But most training datasets Timofte et al. (2017); Li et al. (2023b) focus on collecting higher resolutions and larger quantities of real images, few considering the bias between the training and testing datasets, and there is no dataset taking the synthetic images into account. In this paper, we construct an image complexity metric based on GLCM analysis and find that the commonly used training datasets Timofte et al. (2017); Lim et al. (2017) for SR task and Arbelaez et al. (2010); Ma et al. (2016) for image denoising task, exhibit a certain distribution difference in resolution compared to the testing datasets Bevilacqua et al. (2012); Zeyde et al. (2012); Martin et al. (2001); Huang et al. (2015); Matsui et al. (2017). Therefore, we utilize this metric as a filter criterion and construct a new image ReStoration dataset which including the Real and Synthetic images (ReSyn dataset).

Table 1: Comparison among different datasets.

| Dataset | # Images | Synthetic Images | Multi-data Source | Resolution Range | Tasks |
|---------|----------|------------------|-------------------|------------------|-------|
| DIV2K | 1,000 | × | × | 2K | SR |
| DF2K | 3,650 | × | × | 2K | SR |
| LSDIR | 84,991 | × | × | [2K, 4K] | SR |
| DFWB | 8,805 | × | ✓ | [1K,4K] | Denosing |
| ReSyn (Ours) | 12,000 | ✓ | ✓ | [0.5K,4K] | SR, Denosing, JPEG |

## 2.2 RECEPTANCE WEIGHTED KEY VALUE MODEL (RWKV)

The attention mechanism has shown promising performance in both CV and NLP fields. Various operators with linear complexity Peng et al. (2023); Gu & Dao (2023); Qin et al. (2023) have been explored to optimize the global attention mechanism in recent years. A modified form of linear attention, the Attention Free Transformer (AFT) Zhai et al. (2021), paved the way for the RWKV architecture by using some attention heads equal to the size of the feature dimension and incorporating a set of learned pairwise positional biases. RWKV-v4 Peng et al. (2023) employed exponential decay to model global information efficiently. Vision-RWKV Duan et al. (2024) transfers the RWKV-v4 to the vision domain through a Q-shift mechanism and bidirectional attention. RWKV-5/6 Peng et al. (2024) further refined the architecture of RWKV-4. RWKV-5 adds matrix-valued attention states, LayerNorm over the attention heads, SiLU attention gating, and improved initialization. It also removes the Sigmoid activation of receptance. RWKV-6 further applies data dependence to the decay schedule and token shift. We modify the RWKV-v4 module of Vision-RWKV, use a depth-wised conv to replace the Q-shift, and also a Cross-Bi-Direction attention to replace the Bi-Direction attention.

## 3 RESYN DATASET

Due to limited photography techniques and compression techniques, many images in previous datasets Deng et al. (2009); Timofte et al. (2017) suffer from noise, blurring, and other problems. However, most datasets focus solely on obtaining high-resolution images, using resolution and Bits Per Pixel (BPP) for image filtering, but lacking a sufficient consideration for image complexities. This causes a distribution bias (different image complexities) between the classic image restoration training datasets and the test datasets. We propose an image complexity metric based on the Gray Level Co-occurrence Matrix (GLCM) analysis method to directly analyze this complexity distribution bias. As shown in Fig. 2, we analyze the GLCM complexity distribution of the classic SR training datasets (DIV2K Timofte et al. (2017) and DF2K Lim et al. (2017)), and test datasets (Urban100, Manga109 and BS100). It can be seen that the classic SR training datasets and test datasets often have different image complexity distributions. And the test dataset that often achieves a better PSNR performance, *e.g.,* Manga109, has a complexity distribution closer to that of the training dataset. We further analyze the relationship between the GLCM complexity indicator and the restoration performance metric PSNR. As shown in Fig. 3, GLCM complexity can better predict the restoration performance compared to the BPP indicator. Moreover, datasets with lower GLCM complexity images show better restoration performance.

To enhance the performance of existing methods and incorporate images generated by the AIGC method, we construct the ReSyn dataset, a new large-scale dataset for both the Real and Synthetic image ReStoration. Our dataset introduces the GLCM complexity indicator as a criterion for filtering images to help improve the quality of shuffled images and achieve a balanced complexity distribution. Examples of the images from our ReSyn dataset are shown in Fig.1b. The data collection pipeline and the dataset analysis are introduced in details below. Tab. 1 presents the differences between our ReSyn dataset and commonly used IR training datasets. More details are in the **Appendix** Sec. A.

### 3.1 DATA COLLECTION

**Data Source.** Our dataset consists of both real images and AI-generated images. Many previous methods Chen et al. (2023a); Zhou et al. (2023) improve performance by pretraining on large-scale datasets like ImageNet Deng et al. (2009). We follow this trend and collect images from these datasets to form the real image part of our dataset. The real images are collected from the commonly used large-scale datasets for high-level tasks, including ImageNet Deng et al. (2009), COCO2017 Lin et al. (2014), and SAM Kirillov et al. (2023). The images are filtered and low-quality images are

discarded, only 9K images are retained. To introduce the AI-generated images into this dataset, we automatically crawl images from Midjourney, and after filtering we retain 3K synthetic images for our dataset. The origins of these images are detailed in the distribution chart shown in Fig. 1a, while the diversity of their resolutions is illustrated in Fig. 1b. Different from previous datasets that filter solely based on resolutions, we do not completely discard images with resolutions below 2K, since many images in the test dataset are below 2K resolution and have a high image complexity. Our dataset encompasses a broad range of image resolutions, with images ranging from 0.5K to 4K, and most images have a resolution larger than 1K.

**Data Selection Criteria.** The images used for image restoration model training need to have a high pixel-level quality. To this end, we divide the image filtering process into three steps. 1) Firstly, the images of resolution smaller than $800 \times 800$ are discarded, since for super-resolution tasks, the images need to be down-sampled. This can help remove most low-quality images. 2) Secondly, to remove the blurry or noisy degraded images, we follow the blur and noise suppression process proposed in LSDIR Li et al. (2023b). The remaining images are under blur detection by the variance of the image Laplacian, and flat region detection through the Sobel filter. 3) Thirdly, all the images are shuffled through the GLCM complexity metric (detailed below) to ensure a balanced distribution. We ensure that the number of images with complexity values below zero is equal to that above zero. Therefore, we can form a dataset of balanced image complexity distribution. It should be mentioned that images from different sources are filtered separately.

**Image Complexity Analysis.** As shown in previous works Timofte et al. (2017), the PSNR metric of super-resolution images is strongly correlated with the Bits Per Pixel (BPP), an indicator for the quantity of image information. Although BPP reflects the quality of an image by measuring its color depth, it lacks consideration for the relationships between pixels and cannot adequately measure the texture variation and complexity of an image. Therefore, this paper further measure the image complexity based on the Gray Level Co-occurrence Matrix (GLCM) and investigate its correlation with PSNR metric. Since human eyes are more sensitive to texture, we utilize the GLCM, which is closely related to the complexity of image texture, to construct an image complexity analysis metric. We calculate relevant statistical quantities from the GLCM, and use Entropy, Energy, and Dissimilarity to build a formula for image complexity analysis as follows: $I_{complexity} = ENT - ENE + DISS$, where $ENT$, $ENE$, and $DISS$ represent entropy, energy, and dissimilarity respectively, all of which are statistical quantities calculated from GLCM. As shown in Fig. 3, we analyze the correlation between the PSNR-Y metric and GLCM-based image complexity, as well as BPP. The PSNR metrics are measured on super-resolution results generated by two pre-trained models and a direct bicubic upsample for the Urban100 test images, and sorted according to GLCM complexity and BPP. Our GLCM complexity measure has a stronger Pearson correlation ($\rho = -0.86$) compared to BPP ($\rho = 0.65$), indicating that the proposed image complexity is a stronger predictor for the PSNR metric of the restored images. Furthermore, the distribution of GLCM complexity is symmetric with respect to the origin, making it an excellent metric for image filtering.

**Post Processing.** For the classic image restoration tasks, *e.g.,* super-resolution, the training requires paired down-sampled LR images and ground truths. We employ the classic bicubic down-sampling method[1] to obtain the LR images. We use the commonly used scale factors of $\times 2$, $\times 3$, and $\times 4$.

**Partitions.** After the image filtering and the post-processing, there are 12K images left, of which 9K are real images and 3K are synthetic images. We then randomly partition our ReSyn dataset into a training set of 10K images, a validation set of 1K images, and a test set of 1K images.

## 4 METHODOLOGY: RWKV-IR

The Receptance Weighted Key-Value model (RWKV) Peng et al. (2023) is a linear complexity attention mechanism that combines the advantages of RNN and Transformer. Its linear complexity enables the utilization of a broader range of pixels for activation, which is suitable for image restoration tasks. Consequently, we integrate this linear complexity attention mechanism with a classic image restoration model to construct our **RWKV-IR**. The framework of our RWKV-IR is illustrated in Fig. 4, following the widely used structure Liang et al. (2021); Guo et al. (2024) that

---

[1]Simulating MATLAB's anti-aliasing imresize using a Python-based approach, with negligible differences in visual effects.

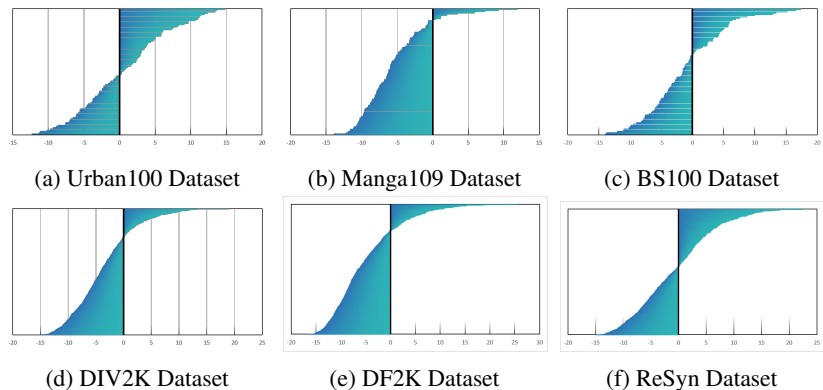

(a) Urban100 Dataset     (b) Manga109 Dataset     (c) BS100 Dataset

(d) DIV2K Dataset     (e) DF2K Dataset     (f) ReSyn Dataset

Figure 2: The complexity distributions of different datasets. The complexity distributions of the training datasets DIV2K Timofte et al. (2017) and DF2K Lim et al. (2017) have a typical shift, containing more images of low complexity. Our ReSyn dataset balances the distribution of low and high complexity images by image filtering based on the newly proposed GLCM image complexity measure.

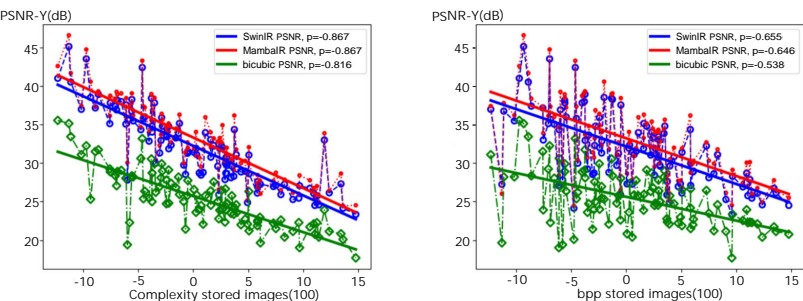

Figure 3: PSNR ($\times 2$ SR on Urban100 Huang et al. (2015)) performance can be predicted by the proposed GLCM image complexity and BPP Timofte et al. (2017). For each predictor, we sort images and compute the Pearson correlation ($\rho$) with PSNR. Compared to BPP, GLCM Complexity has a higher correlation to PSNR.

consists of three stages: shallow feature extraction, deep feature enhancing, and high-quality image reconstruction, where the RWKV is mainly incorporated into the second stage. 1) Shallow feature extraction: Given a low-quality input $I_{LQ} \in \mathbb{R}^{H \times W \times 3}$, a $3 \times 3$ convolution layer first extracts the shallow feature $F_S \in \mathbb{R}^{H \times W \times C}$, where $H$ and $W$ represent the height and width, and $C$ is the number of channels in the shallow feature. 2) Deep feature enhancing: Subsequently, a series of Global&Local Linear attention Layers (GLLL, which is based on RWKV) and a $3 \times 3$ Convolution Block perform deep feature extraction. Each GLLL layer contains several GLLB blocks, each of which consists of a Global&Local-Aware Spatial Mix module and a Channel Mix module. Afterwards, a global residual connection fuses the shallow feature $F_S$ and the deep feature $F_D$ into a hybrid feature $F_H = F_S + F_D$, which is then input into the high-quality image reconstruction module. 3) High-quality image reconstruction: Finally, the high-quality reconstruction module outputs a restored image $I_{RE} \in \mathbb{R}^{(H \times s) \times (W \times s) \times 3}$, where $s$ is the scale factor used for the super-resolution task.

**Global&Local Linear attention Block (GLLB).** Transformer-based restoration networks Liang et al. (2021); Chen et al. (2023a) typically design the core block for restoration following a "Norm $\rightarrow$ Attention $\rightarrow$ Norm $\rightarrow$ MLP" workflow. The Attention module is designed to model global dependency, but due to heavy computational complexity, only local window attention is used. Therefore, replacing the local attention with a linear complexity attention mechanism can reduce the computational overhead while increasing the window size, to better model global dependencies. Therefore, we aim to incorporate the linear complexity Receptance Weighted Key Value (RWKV) mechanism to enhance the image restoration effects. However, simply replacing the Attention module with the Spatial Mix from RWKV, and replacing the MLP module with the Channel Mix from RWKV

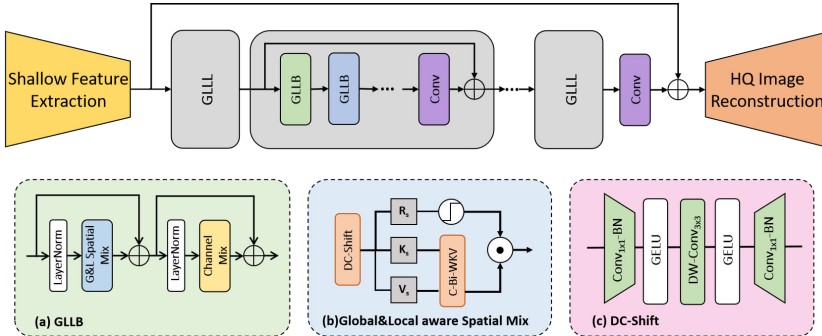

Figure 4: **Framework of our RWKV-IR**, which consists of three stages: shallow feature extraction, deep feature enhancing, and HQ image reconstruction. For deep feature enhancing, a series of Global&Local Linear attention Layers (GLLL, which is based on RWKV) and a Conv Block are used. Each GLLL layer contains several GLLB blocks, each of which contain a Global&Local-Aware Spatial Mix module and a Channel Mix module.

yields sub-optimal results (as shown in Tab. 3, 4). We find that the linear complexity attention module from RWKV can model global dependencies well, but its utilization of local information is insufficient. Through further experiments, we find that this is caused by the Q-shift and the Bi-direction WKV in the original RWKV, which are transferred from the NLP tasks and not suitable for the low-level vision tasks. Therefore, we replace the Q-shift with a newly proposed **Depth-wise Convolution shift (DC-shift)** to achieve better visual representation and easier optimization Yuan et al. (2021); Zhao et al. (2021); Chen et al. (2023a). Moreover, we propose a **Cross-Bi-WKV module** that integrates two Bi-WKV modules with different scanning orders, instead of the original Bi-direction WKV, to balance the attention for horizontal and vertical directions and improve the model performance.

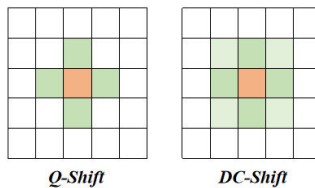

Figure 5: Different shift methods. The Q-shift is a simple channel replacement operation using four neighboring pixels, while our DC-shift is a depth-wise conv leveraging the surrounding pixels in a $k \times k$ neighborhood.

As shown in Fig. 4(a), we propose a **Global**&**Local Linear attention Block (GLLB)**, which follows a "Norm → Conv → Attention → Norm → Channel-Mix" workflow, with two residual connections. Given an intermediate feature $F_i$, where $i$ represents the $i$-th GLLB block. A LayerNorm module is followed by a linear complexity **Global**&**Local-Aware Spatial Mix** module, that models both long-term dependencies and local dependencies: $F_{g,i} = \text{GLSpatial-Mix}(\text{LN}(F_i))$. The local-aware characteristics are achieved by replacing the Q-Shift in the original Spatial Mix with our Depth-wise Convolution shift (DC-Shift, detailed below). Furthermore, the Channel Mix module replacing the MLP is used for stabilizing the training process and avoiding channel oblivion: $F_{i+1} = F_{g,i} * \beta + \text{LN}(\text{Channel-Mix}(F_{g,i}))$, where $\beta$ is a learnable parameter.

**DC-Shift.** To emulate the memory mechanism of RNNs, the original RWKV proposes a token shift mechanism. Consider an input feature $X \in \mathbb{R}^{T \times C}$ (where $T = H \times W$), it is first shifted, and then fed into three linear layers to obtain the matrices $R_s, K_s, V_s \in \mathbb{R}^{T \times C}$:

$$R_s = \text{Shift}_R(X)W_R, \quad K_s = \text{Shift}_K(X)W_K, \quad V_s = \text{Shift}_V(X)W_V. \tag{1}$$

Then, $K_s$ and $V_s$ are used to calculate the global attention $wkv \in \mathbb{R}^{T \times C}$ by a linear complexity bidirectional attention mechanism, and multiplied with $\sigma(R_s)$ which controls the output $O_s$ probability. But the original implementation of Shift is a Q-shift operation, which simply combines the features from the top, left, down, and right neighboring pixels, each using $C/4$ channels, to replace the feature of the center pixel, formulated as follows:

$$\text{Q-Shift}_{(*)}(X) = X + (1 + \mu_{(*)})X',$$
$$where \quad X'[h, w] = \text{Concat}(X[h-1, w, 0:C/4], X[h+1, w, C/4:C/2], \tag{2}$$
$$X[h, w-1, C/2:3C/4], X[h, w+1, 3C/4:C]).$$

The Q-Shift is not suitable for image restoration due to two reasons: 1) In image restoration tasks, the number of channels in the features is relatively small compared to NLP tasks; and 2) The simple feature substitution in Q-Shift does not consider the similarity between local pixels, making it not suitable for image restoration tasks that rely on local similarity. Therefore, we propose a **Depth-wise Convolution Shift (DC-Shift)** shown in Fig. 5 to replace the Q-shift, which helps enhance the model performance by modeling the relationships in local receptive fields. As shown in Fig. 4(c), the DC-Shift consists of two $1\times1$ Convolution Layers and one $ks \times ks$ Depth-wise Convolution Layer. By using this structure, we can reduce the number of parameters compared to using the classic channel convolution module and also compensate for the lack of local features. The calculation process of this DC-Shift module is formulated as: $F_{l,i} = \text{Conv}_{1\times1}(\text{GeLU}(\text{DW-Conv}_{ks\times ks}(\text{GeLU}(\text{Conv}_{1\times1}(X)))))$, where $ks$ is the kernel size of the depth-wise convolution. By combining the Depth-wise Convolution Shift with Bi-directional attention, as shown in Fig. 4(b), we achieve Global&Local-Aware Spatial Mix.

**Cross-Bi-WKV module.** The core idea of the Vision-RWKV is the linear complexity Bi-directional attention (Bi-WKV). Its calculation result for the $t$-th pixel is formulated as:

$$wkv_t = \text{Bi-WKV}(K, V)_t = \frac{\sum_{i=0,i\neq t}^{T-1} e^{-(|t-i|-1)/T\dot{w}+k_i}v_i + e^{u+k_t}v_t}{\sum_{i=0,i\neq t}^{T-1} e^{-(|t-i|-1)/T\dot{w}+k_i}v_i + e^{u+k_t}}, \quad (3)$$

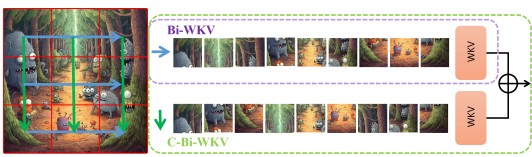

Figure 6: Illustration of Cross-Bi-WKV, which consists of two cross scanning Bi-WKV modules.

where the upper limit for the current pixel $t$ (the $t-th$ pixel after flattening the 2D pixels into a 1D sequence using the horizontal scaning order) is set to $T-1$ (the last pixel), to ensure that all pixels are mutually visible in the calculation of each other's result. In this formula, $(|t-i|-1)/T$ is used as the position embedding, which is unbalanced for horizontal and vertical directions, *i.e.,* the position embedding differences between the left and right neighboring pixels are much smaller than that between the up and down neighboring pixels, and after applying the negative sign and exponential calculation, this leads to more attention to the horizontal direction than the vertical direction. Therefore, we propose a **Cross-Bi-directional WKV module**, which combines a horizontal direction Bi-WKV and a vertical direction Bi-WKV to achieve a balanced attention to horizontal and vertical pixels. The two Bi-WKV modules use **different scanning orders** when flattening the pixels into a 1D sequence, one using the horizontal scanning order, and the other using the vertical scanning order, as shown in Fig. 6. And we use the average output of the two Bi-WKV modules to form the final output feature. After the DC-Shift, our Cross-Bi-directional attention mechanism with a linear complexity is formulated as follows and outputs a global attention result $wkv \in \mathbb{R}^{T \times C}$:

$$wkv = \text{C-Bi-WKV}(K, V) = \text{Bi-WKV}_{horizontal}(K_h, V_h) + \text{Bi-WKV}_{vertical}(K_v, V_v). \quad (4)$$

The result $wkv$ is then multiplied with $\sigma(R)$ to obtain the output $O_s$ probability: $O_s = (\sigma(R_s) \bigodot wkv)W_O$. As the RWKV model continues to iterate and update, we believe that subsequent versions will bring greater improvements.

## 5 EXPERIMENTS

Few previous methods have focused on the impact of training criteria on the performance comparison of restoration models. Many approaches Liang et al. (2021); Zhou et al. (2023); Guo et al. (2024) have instead opted to extend training time continuously to improve model performance. This can lead to unfair comparisons in subsequent evaluations. Therefore, we construct a comprehensive training benchmark from two perspectives: measuring the model's convergence ability and its restoration capability. In this section, we present the experimental benchmark and the training details of our proposed linear-complexity attention-based image restoration model. We then conduct a benchmark study comparing our method with other models. *Due to space limitations, only the results of classical SR tasks are shown in the main paper. Please refer to the **Appendix** Sec. C for the experiments of other IR tasks (light-weight SR, image denoising, JPEG artifacts reduction). Source codes are provided in the supplementary material.*

## 5.1 EXPERIMENTAL SETTINGS

**Experimental Benchmark.** We propose a unified training benchmark for different kinds of image restoration tasks. *1) Super Resolution:* The commonly conducted SR tasks are lightweight SR and classical SR. For all compared methods, we set the same batch size and the same number of iterations for training. To show models' convergence and restoration abilities, we compared different models in two number levels of training iterations. The batch size for lightweight SR model training is 64, and the training iterations are 50K and 500K. The batch size for classical SR model training is 32, and the training iterations are 100K and 500K. *2) Image Denoising:* For Gaussian color denoising and gray-scale denoising tasks, the training batch size and the training iterations are set to 16 and 100K (500k for long training), respectively. *3) JPEG Compression Artifact Reduction:* the batch size and training iterations are set to 16 and 100K (500k for high number level), respectively. *For more details about the model settings, please refer to the Appendix Sec. B.*

**Training Details.** We conduct super-resolution (SR) training experiments on three datasets: DIV2K Timofte et al. (2017), DF2K Lim et al. (2017), and our ReSyn. For lightweight SR, models are trained separately on the DIV2K and ReSyn datasets. For classic SR, models are trained separately on the DF2K and ReSyn datasets. In the training of image denoising models, we compare performance on the DFWB RGB dataset (a combined dataset of DIV2K Wang et al. (2023), Flickr2K Lim et al. (2017), BSD500 Arbelaez et al. (2010), and WED Ma et al. (2016)) and our ReSyn dataset. During training, we crop the compressed images into $64 \times 64$ patches for image SR. We do not use pre-trained weights from the $\times 2$ model to initialize those of $\times 3$ and $\times 4$ but train the $\times 3$ and $\times 4$ models from scratch. For the denoising task, we crop the original images into $128 \times 128$ patches. We employ the Adam optimizer for training our RWKV-IR with $\beta_1 = 0.9$ and $\beta_2 = 0.999$. The initial learning rate is set at $2 \times 10^{-4}$ and is decreased during training using the multi-step scheduler. Our models are trained with 8 NVIDIA V100 GPUs. Except for the batch size and training iterations for the compared methods, all other settings remain consistent with their official training codes.

## 5.2 COMPARISON ON IMAGE SUPER-RESOLUTION

**Classic Image Super-Resolution.** Table 2 presents quantitative comparisons between RWKV-IR and state-of-the-art methods (HAN Niu et al. (2020), SwinIR Liang et al. (2021), SRFormer Zhou et al. (2023), and MambaIR Guo et al. (2024)) on 100K training iters, which can show the convergence capability of models. Our method achieves optimal results on almost all five datasets for all scale factors. As shown, our RWKV-based baseline outperforms SwinIR by 0.08dB on Urban100 for x4 scale and MambaIR by 0.03dB, demonstrating the image restoration capability and quick convergence ability of our RWKV-IR. Furthermore, training our newly constructed ReSyn dataset also achieves decent performance, despite the overall quality of the data sources not being high. This also proves the feasibility of the image complexity-based dataset construction method. *The comparisons on long training iterations and other IR tasks could be found in the Appendix Sec. C.*

## 5.3 ABLATION STUDY

**Effects of different designs of GLLB.** In this section, we conduct ablation studies to explore the effects of different designs of the core GLLB on the test dataset Urban100. These experiments can demonstrate the issues that need to be considered when applying the linear attention mechanism RWKV to image restoration models. They also provide an intuitive reflection of the problems mentioned earlier, offering insights for subsequent application research. To reduce training costs, all the models are lightweight models trained on the DIV2K dataset. The ablation studies results in Tab. 3, 4 indicate that: (1) The original Q-Shift from the RWKV hinders the performance of the model on the image restoration task, since its simple feature substitution does not capture local similarity. (2) We propose the Depth-wise Convolution Shift (DC-Shift) to replace the Q-shift, which models relationships in local receptive fields, helping to enhance the restoration capabilities, thereby obtaining a PSNR improvement of 0.87dB. (3) Since the Bi-WKV pays unbalance attention to horizontal and vertical directions, our Cross Bi-WKV module that combines two Bi-WKV modules with different scanning orders further improves the performance of the image restoration model, with a PSNR improvement of 0.75dB. (4) The MLP module is not suitable for image restoration, a Channel Attention Block (CAB) or Channel Mix process can further improve the model performance.

Table 2: Quantitative comparison on **classic image super-resolution** with state-of-the-art methods on 10K iters training. The best and the second best results are in red and blue.

| Method | scale | dataset | Set5 | | Set14 | | BSDS100 | | Urban100 | | Manga109 | | ReSyn | |
|---|---|---|---|---|---|---|---|---|---|---|---|---|---|---|
| | | | PSNR | SSIM | PSNR | SSIM | PSNR | SSIM | PSNR | SSIM | PSNR | SSIM | PSNR | SSIM |
| HAN Niu et al. (2020) | ×2 | DF2K | 38.26 | 0.9611 | 34.01 | 0.9205 | 32.36 | 0.9008 | 33.09 | 0.9366 | 39.56 | 0.9788 | 35.22 | 0.9311 |
| SwinIR Liang et al. (2021) | ×2 | DF2K | 38.25 | 0.9616 | 34.04 | 0.9215 | 32.39 | 0.9024 | 33.06 | 0.9365 | 39.54 | 0.9790 | 35.24 | 0.9313 |
| SRFormer Zhou et al. (2023) | ×2 | DF2K | 34.80 | 0.9301 | 30.74 | 0.8495 | 29.33 | 0.8113 | 29.12 | 0.8712 | 34.68 | 0.9510 | 32.30 | 0.8763 |
| MambaIR Guo et al. (2024) | ×2 | DF2K | 38.34 | 0.9617 | 34.42 | 0.9246 | 32.45 | 0.9032 | 33.55 | 0.9401 | 39.78 | 0.9796 | 35.40 | 0.9325 |
| RWKV-IR (Ours) | ×2 | DF2K | 38.38 | 0.9618 | 34.42 | 0.9246 | 32.46 | 0.9034 | 33.59 | 0.9404 | 39.80 | 0.9797 | 35.44 | 0.9327 |
| HAN Niu et al. (2020) | ×2 | ReSyn | 38.08 | 0.9002 | 33.88 | 0.9198 | 32.30 | 0.9011 | 33.10 | 0.9352 | 39.15 | 0.9733 | 35.46 | 0.9328 |
| SwinIR Liang et al. (2021) | ×2 | ReSyn | 38.18 | 0.9616 | 34.03 | 0.9213 | 32.39 | 0.9026 | 33.12 | 0.9370 | 39.30 | 0.9791 | 35.52 | 0.9331 |
| SRFormer Zhou et al. (2023) | ×2 | ReSyn | 38.17 | 0.9621 | 34.12 | 0.9224 | 32.41 | 0.9031 | 33.39 | 0.9396 | 39.50 | 0.9797 | 35.62 | 0.9340 |
| MambaIR Guo et al. (2024) | ×2 | ReSyn | 38.26 | 0.9616 | 34.43 | 0.9247 | 32.46 | 0.9034 | 33.54 | 0.9401 | 39.72 | 0.9794 | 35.64 | 0.9350 |
| RWKV-IR (Ours) | ×2 | ReSyn | 38.28 | 0.9618 | 34.42 | 0.9245 | 32.47 | 0.9032 | 33.58 | 0.9404 | 39.76 | 0.9796 | 35.68 | 0.9352 |
| HAN Niu et al. (2020) | ×3 | DF2K | 34.77 | 0.9300 | 30.60 | 0.8466 | 29.28 | 0.8110 | 29.03 | 0.8701 | 32.56 | 0.9451 | 32.25 | 0.8744 |
| SwinIR Liang et al. (2021) | ×3 | DF2K | 34.80 | 0.9301 | 30.74 | 0.8495 | 29.33 | 0.8113 | 29.12 | 0.8712 | 34.65 | 0.9501 | 32.30 | 0.8763 |
| SRformer Zhou et al. (2023) | ×3 | DF2K | 34.62 | 0.9306 | 30.73 | 0.8501 | 29.39 | 0.8132 | 29.53 | 0.8784 | 34.92 | 0.9524 | 32.38 | 0.8779 |
| MambaIR Guo et al. (2024) | ×3 | DF2K | 34.86 | 0.9307 | 30.76 | 0.8505 | 29.39 | 0.8123 | 29.42 | 0.8758 | 34.92 | 0.9519 | 32.42 | 0.8780 |
| RWKV-IR (Ours) | ×3 | DF2K | 34.88 | 0.9308 | 30.77 | 0.8507 | 29.40 | 0.8124 | 29.43 | 0.8757 | 34.95 | 0.9520 | 32.43 | 0.8782 |
| HAN Niu et al. (2020) | ×3 | ReSyn | 34.49 | 0.9278 | 30.36 | 0.8427 | 29.20 | 0.8110 | 28.73 | 0.8711 | 34.40 | 0.9500 | 32.38 | 0.8772 |
| SwinIR Liang et al. (2021) | ×3 | ReSyn | 34.68 | 0.9296 | 30.71 | 0.8487 | 29.30 | 0.8113 | 29.20 | 0.8723 | 34.44 | 0.9505 | 32.41 | 0.8776 |
| SRFormer Zhou et al. (2023) | ×3 | ReSyn | 34.48 | 0.9298 | 30.72 | 0.8488 | 29.34 | 0.8127 | 29.46 | 0.8771 | 34.53 | 0.9512 | 32.47 | 0.8786 |
| MambaIR Guo et al. (2024) | ×3 | ReSyn | 34.74 | 0.9298 | 30.74 | 0.8492 | 29.35 | 0.8122 | 29.42 | 0.8766 | 34.53 | 0.9511 | 32.49 | 0.8803 |
| RWKV-IR (Ours) | ×3 | ReSyn | 34.77 | 0.9299 | 30.73 | 0.8491 | 29.34 | 0.8123 | 29.45 | 0.8768 | 34.55 | 0.9513 | 32.50 | 0.8805 |
| HAN Niu et al. (2020) | ×4 | DF2K | 32.51 | 0.9001 | 28.85 | 0.7856 | 27.75 | 0.7440 | 26.88 | 0.8155 | 31.72 | 0.9015 | 30.60 | 0.8321 |
| SwinIR Liang et al. (2021) | ×4 | DF2K | 32.74 | 0.9020 | 29.02 | 0.7920 | 27.83 | 0.7457 | 27.12 | 0.8162 | 31.75 | 0.9223 | 30.67 | 0.8335 |
| SRFormer Zhou et al. (2023) | ×4 | DF2K | 32.68 | 0.9010 | 29.03 | 0.7917 | 27.82 | 0.7458 | 27.38 | 0.8218 | 31.86 | 0.9234 | 30.72 | 0.8346 |
| MambaIR Guo et al. (2024) | ×4 | DF2K | 32.73 | 0.9015 | 29.05 | 0.7923 | 27.87 | 0.7465 | 27.20 | 0.8176 | 31.83 | 0.9228 | 30.73 | 0.8345 |
| RWKV-IR (Ours) | ×4 | DF2K | 32.74 | 0.9016 | 29.04 | 0.7924 | 27.86 | 0.7466 | 27.21 | 0.8177 | 31.87 | 0.9233 | 30.74 | 0.8350 |
| HAN Niu et al. (2020) | ×4 | ReSyn | 32.29 | 0.8991 | 28.59 | 0.7878 | 27.64 | 0.7420 | 26.58 | 0.8155 | 31.55 | 0.9201 | 30.67 | 0.8321 |
| SwinIR Liang et al. (2021) | ×4 | ReSyn | 32.63 | 0.9009 | 28.98 | 0.7908 | 27.81 | 0.7449 | 27.16 | 0.8158 | 31.60 | 0.9220 | 30.75 | 0.8342 |
| SRFormer Zhou et al. (2023) | ×4 | ReSyn | 32.61 | 0.9007 | 28.98 | 0.7909 | 27.82 | 0.7452 | 27.42 | 0.8220 | 31.72 | 0.9225 | 30.79 | 0.8350 |
| MambaIR Guo et al. (2024) | ×4 | ReSyn | 32.65 | 0.9009 | 29.00 | 0.7910 | 27.85 | 0.7458 | 27.21 | 0.8181 | 31.79 | 0.9224 | 30.81 | 0.8350 |
| RWKV-IR (Ours) | ×4 | ReSyn | 32.66 | 0.9011 | 29.02 | 0.7911 | 27.86 | 0.7459 | 27.24 | 0.8183 | 31.81 | 0.9226 | 30.83 | 0.8351 |

These experiments show the Cross-Bi-WKV module and the Depth-wise Conv Shift can help improve the performance. With the continuous iteration and upgrades of RWKV, we believe that subsequent versions of RWKV will bring genuine global attention, significantly enhancing the IR capabilities.

| DC-Shift Position | Before SM | Between SM and CM | Behind CM | Parallel | **Replace QS** | | WKV Setting | Bi-WKV | **Cross-Bi-WKV** |
|---|---|---|---|---|---|---|---|---|---|
| PSNR ↑ | 32.41 | 32.54 | 32.51 | 32.65 | 32.95 | | PSNR ↑ | 32.20 | 32.95 |

Table 3: Ablation study of DC-Shift insertion position and different WKV methods. Left: the study of insertion position. SM, CM, and QS are Spatial Mix, Channel Mix, and Q-shift modules respectively. Right: the study of the different settings of WKV scanning methods. Ours settings of are in **Bold**.

| Shift Method | Q-Shift (p=1) | Q-Shift (p=0, w.o. shift) | **DC-Shift (ks=3)** | DC-Shift (ks=5) | | FFN | MLP | CAB | **Channel Mix** |
|---|---|---|---|---|---|---|---|---|---|
| PSNR ↑ | 32.08 | 32.69 | 32.95 | 32.84 | | PSNR ↑ | 32.32 | 32.67 | 32.95 |

Table 4: Ablation study of the different settings of Shift methods, and the feed forward layer (FFN) after the attention module. Left: the study of Shift settings ($p$ in Q-Shift is the distance from neighboring pixels to the center; $ks$ in DC-Shift is the kernel size). Right: the study of FFN modules (CAB means Channel Attention Block). Ours settings of are in **Bold**. The best scores are in red.

# 6 CONCLUSION

In this paper, we review the task of image restoration. We utilize the Gray Level Co-occurrence Matrix to construct an image complexity metric, which demonstrates the bias between complexity distributions of classic training datasets and test datasets. Based on this metric, we construct ReSyn, a new image restoration dataset that includes both real and generated images with balanced complexity. Additionally, we develop a novel benchmark for comparing image restoration models, focusing on two aspects: the convergence speed and the restoration capability. Moreover, from the perspective of linear attention mechanisms, we propose a novel RWKV-IR model, which integrates the RWKV into image restoration models, constructing a linear attention-based image restoration model. This inspires further exploration and enhancement of the effective receptive field of the model. Extensive experiments demonstrate the effectiveness of our proposed benchmark and RWKV-IR model.

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

APPENDIX

## A  MORE DETAILS OF OUR RESYN DATASET

In this section, we further provide details on the construction of our ReSyn dataset. The process of final filtering based on our GLCM image complexity metric is shown in Fig. 7.

Our ReSyn dataset contains images from four sources, including ImageNet Deng et al. (2009), COCO2017 Lin et al. (2014), SAM Kirillov et al. (2023), and MidJourney. For ImageNet, we first remove images with a resolution of less than 800×800. Since the details in the images from ImageNet are not rich and relatively blurry, then the images of blurring and noise degradation are filtered. Finally, through image complexity filtering, we choose 1,200 images from the ImageNet. Half of the images have a complexity greater than zero.

For COCO2017 Lin et al. (2014), almost all of the images are medium-resolution images, we follow the method of ImageNet filtering and include 1,200 images from COCO2017.

For SAM, most images have a resolution of over 2K. For privacy purposes, many of the images in the dataset containing faces and sensitive information are mosaiced. Therefore, we manually remove the images that include the mosaics, leaving only the clear images. After this step, we use the same filtering method as ImageNet and include 6,000 images from SAM.

For MidJourney, we first crawl more than 30,000 high-quality images from the web. And then after filtering, 3,600 images are left to form the dataset.

## B  MODEL DETAILS

In Tab. 5, we provide the model setting details for different image restoration tasks, which could serve as a reference for model construction. It should be noted that the number of embedding channels in RWKV-based models must be an integer that is a multiple of 16.

## C  MORE IMAGE RESTORATION EXPERIMENTS

In this section we supplement quantitative comparisons on other image restoration tasks, including 1) light-weight SR, 2) image denoising, and 3) JPEG artifacts reduction. These experiments show the generality of our model.

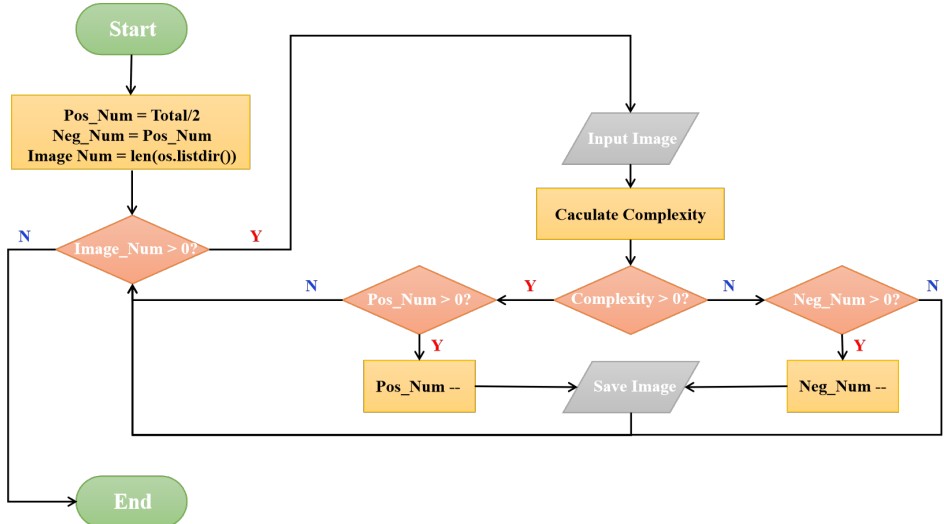

Figure 7: The final filtering procedure based on GLCM image complexity of the ReSyn dataset.

Table 5: The model setting details for different image restoration tasks.

| | Classic SR | Light-Weight SR | Denoising | JPEG |
|---|---|---|---|---|
| Embed channel | 192 | 48 | 192 | 192 |
| Image size | 64 | 64 | 128 | 128 |
| Blocks setting | [6,6,6,6,6,6] | [6,6,6,6] | [6,6,6,6,6,6] | [6,6,6,6,6,6] |
| WKV setting | Cross WKV | Layer Cross WKV | Cross WKV | Cross WKV |

Table 6: Quantitative comparison on **classic image super-resolution** with state-of-the-art methods on 500K training iterations.

| Method | scale | Set5 PSNR | Set5 SSIM | Set14 PSNR | Set14 SSIM | BSDS100 PSNR | BSDS100 SSIM | Urban100 PSNR | Urban100 SSIM | Manga109 PSNR | Manga109 SSIM |
|---|---|---|---|---|---|---|---|---|---|---|---|
| EDSR Lim et al. (2017) | ×2 | 38.11 | 0.9602 | 33.92 | 0.9195 | 32.32 | 0.9013 | 32.93 | 0.9351 | 39.10 | 0.9773 |
| SAN Dai et al. (2019) | ×2 | 38.31 | 0.9620 | 34.07 | 0.9213 | 32.42 | 0.9028 | 33.10 | 0.9370 | 39.32 | 0.9792 |
| HAN Niu et al. (2020) | ×2 | 38.27 | 0.9614 | 34.16 | 0.9217 | 32.41 | 0.9027 | 33.35 | 0.9385 | 39.46 | 0.9785 |
| ELAN Zhang et al. (2022b) | ×2 | 38.36 | 0.9620 | 34.20 | 0.9228 | 32.45 | 0.9030 | 33.44 | 0.9391 | 39.62 | 0.9793 |
| SwinIR Liang et al. (2021) | ×2 | 38.42 | 0.9623 | 34.46 | 0.9250 | 32.53 | 0.9041 | 33.81 | 0.9427 | 39.92 | 0.9797 |
| SRFormer Zhou et al. (2023) | ×2 | 38.51 | 0.9627 | 34.44 | 0.9253 | 32.57 | 0.9046 | 34.09 | 0.9449 | 40.07 | 0.9802 |
| MambaIR Guo et al. (2024) | ×2 | 38.60 | 0.9628 | 34.69 | 0.9260 | 32.60 | 0.9048 | 34.17 | 0.9443 | 40.33 | 0.9806 |
| RWKV-IR (Ours) | ×2 | 38.62 | 0.9629 | 34.63 | 0.9254 | 32.58 | 0.9045 | 34.20 | 0.9446 | 40.34 | 0.9804 |
| EDSR Lim et al. (2017) | ×3 | 34.65 | 0.9280 | 30.52 | 0.8462 | 29.25 | 0.8093 | 28.80 | 0.8653 | 34.17 | 0.9476 |
| SAN Dai et al. (2019) | ×3 | 34.75 | 0.9300 | 30.59 | 0.8476 | 29.33 | 0.8112 | 28.93 | 0.8671 | 34.30 | 0.9494 |
| HAN Niu et al. (2020) | ×3 | 34.75 | 0.9299 | 30.67 | 0.8483 | 29.32 | 0.8110 | 29.10 | 0.8705 | 34.48 | 0.9500 |
| ELAN Zhang et al. (2022b) | ×3 | 34.90 | 0.9313 | 30.80 | 0.8504 | 29.38 | 0.8124 | 29.32 | 0.8745 | 34.73 | 0.9517 |
| SwinIR Liang et al. (2021) | ×3 | 34.97 | 0.9318 | 30.93 | 0.8534 | 29.46 | 0.8145 | 29.75 | 0.8826 | 35.12 | 0.9537 |
| SRformer Zhou et al. (2023) | ×3 | 35.02 | 0.9323 | 30.94 | 0.8540 | 29.48 | 0.8156 | 30.04 | 0.8865 | 35.26 | 0.9543 |
| MambaIR Guo et al. (2024) | ×3 | 35.13 | 0.9326 | 31.06 | 0.8541 | 29.53 | 0.8162 | 29.98 | 0.8838 | 35.55 | 0.9549 |
| RWKV-IR (Ours) | ×3 | 35.16 | 0.9328 | 31.02 | 0.8538 | 29.52 | 0.8159 | 30.02 | 0.8839 | 35.57 | 0.9548 |
| EDSR Lim et al. (2017) | ×4 | 32.46 | 0.8968 | 28.80 | 0.7876 | 27.71 | 0.7420 | 26.64 | 0.8033 | 31.02 | 0.9148 |
| SAN Dai et al. (2019) | ×4 | 32.64 | 0.9003 | 28.92 | 0.7888 | 27.78 | 0.7436 | 26.79 | 0.8068 | 31.18 | 0.9169 |
| HAN Niu et al. (2020) | ×4 | 32.64 | 0.9002 | 28.90 | 0.7890 | 27.80 | 0.7442 | 26.85 | 0.8094 | 31.42 | 0.9177 |
| ELAN Zhang et al. (2022b) | ×4 | 32.75 | 0.9022 | 28.96 | 0.7914 | 27.83 | 0.7459 | 27.13 | 0.8167 | 31.68 | 0.9226 |
| SwinIR Liang et al. (2021) | ×4 | 32.92 | 0.9044 | 29.09 | 0.7950 | 27.92 | 0.7489 | 27.45 | 0.8254 | 32.03 | 0.9260 |
| SRFormer Zhou et al. (2023) | ×4 | 32.93 | 0.9041 | 29.08 | 0.7953 | 27.94 | 0.7502 | 27.68 | 0.8311 | 32.21 | 0.9271 |
| MambaIR Guo et al. (2024) | ×4 | 33.13 | 0.9054 | 29.25 | 0.7971 | 28.01 | 0.7510 | 27.80 | 0.8303 | 32.48 | 0.9281 |
| RWKV-IR (Ours) | ×4 | 33.14 | 0.9056 | 29.20 | 0.7968 | 27.99 | 0.7511 | 27.83 | 0.8305 | 32.51 | 0.9285 |

## C.1 CLASSICAL IMAGE SUPER-RESOLUTION

In Tab. 6, we compare RWKV-IR with other methods on 500K training iterations. Our newly proposed image restoration models also have a good performance once the training iteration is long. It also has a linear computational complexity, which makes the model save the computational overhead and is more conducive to the scaling of the model.

## C.2 LIGHTWEIGHT IMAGE SUPER-RESOLUTION

We also provide comparison of our RWKV-IR-light with state-of-the-art lightweight image SR methods: SwinIR Liang et al. (2021), SRFormer Zhou et al. (2023) and MambaIR Guo et al. (2024). Including PSNR and SSIM, we also compare the total number of parameters and MACs (multiply-accumulate operations) to show the model size and the computational complexity of different models. We compare the metrics gained from different training datasets used. The results on 50K training iterations are shown in Tab. 7 and on 500K are shown in Tab. 8. On the small number of training iterations, RWKV-IR outperforms MambaIR-light by 0.10dB on Urban100 with an x4 scale, with a similar parameter number and MACs when trained on the DIV2K dataset. Using ReSyn gets a similar result with DIV2K, since the lightweight models' training iteration is small, our ReSyn could also show a favorable performance. On the large number of training iterations, our proposed linear-complexity attention-based image restoration model also shows competitive performance. This indicates that our model not only converges quickly, but also has excellent image restoration capabilities.

Table 7: Quantitative comparison on **lightweight image super-resolution** with state-of-the-art methods on 50K training iterations. The best scores are in red.

| Method | scale | #param | MACs | dataset | Set5 | | Set14 | | BSDS100 | | Urban100 | | Manga109 | | ReSyn | |
|---|---|---|---|---|---|---|---|---|---|---|---|---|---|---|---|---|
| | | | | | PSNR | SSIM | PSNR | SSIM | PSNR | SSIM | PSNR | SSIM | PSNR | SSIM | PSNR | SSIM |
| SwinIR-light Liang et al. (2021) | ×2 | 878K | 195.6G | DIV2K | 37.76 | 0.9598 | 33.34 | 0.9161 | 32.05 | 0.8981 | 31.55 | 0.9225 | 38.08 | 0.9759 | 34.81 | 0.9276 |
| SRFormer-light Zhou et al. (2023) | ×2 | 853K | 236G | DIV2K | 37.68 | 0.9593 | 33.29 | 0.915 | 32.03 | 0.8977 | 31.56 | 0.9224 | 38.11 | 0.9763 | 34.93 | 0.9279 |
| MambaIR Guo et al. (2024) | ×2 | 859K | 198.1G | DIV2K | 37.88 | 0.9601 | 33.50 | 0.9168 | 32.13 | 0.8993 | 31.97 | 0.9268 | 38.50 | 0.9767 | 34.96 | 0.9288 |
| RWKV-IR (Ours) | ×2 | 863K | 198.5G | DIV2K | 37.98 | 0.9604 | 33.51 | 0.9167 | 32.13 | 0.8991 | 32.15 | 0.9283 | 38.66 | 0.9769 | 34.96 | 0.9287 |
| SwinIR-light Liang et al. (2021) | ×2 | 878K | 195.6G | ReSyn | 37.62 | 0.9594 | 33.29 | 0.9155 | 32.04 | 0.8980 | 31.54 | 0.9224 | 37.87 | 0.9760 | 34.92 | 0.9280 |
| SRFormer-light Zhou et al. (2023) | ×2 | 853K | 236G | ReSyn | 37.62 | 0.9594 | 33.29 | 0.9155 | 32.04 | 0.8980 | 31.54 | 0.9224 | 37.87 | 0.9760 | 34.92 | 0.9280 |
| MambaIR Guo et al. (2024) | ×2 | 859K | 198.1G | ReSyn | 37.68 | 0.9596 | 33.43 | 0.9162 | 32.10 | 0.8989 | 31.80 | 0.9251 | 38.52 | 0.9770 | 35.06 | 0.9290 |
| RWKV-IR (Ours) | ×2 | 863K | 198.5G | ReSyn | 37.79 | 0.9601 | 33.36 | 0.9165 | 32.09 | 0.8990 | 31.98 | 0.9263 | 38.65 | 0.9775 | 35.11 | 0.9295 |
| SwinIR-light Liang et al. (2021) | ×3 | 886K | 87.2G | DIV2K | 34.11 | 0.9246 | 30.22 | 0.8399 | 28.97 | 0.8023 | 27.71 | 0.8426 | 33.00 | 0.9408 | 31.79 | 0.8687 |
| SRFormer-light Zhou et al. (2023) | ×3 | 861K | 105G | DIV2K | 34.24 | 0.9259 | 30.26 | 0.8408 | 29.01 | 0.8036 | 27.88 | 0.8474 | 33.21 | 0.9424 | 31.85 | 0.8699 |
| MambaIR Guo et al. (2024) | ×3 | 867K | 88.7G | DIV2K | 34.32 | 0.9263 | 30.27 | 0.8406 | 29.05 | 0.8039 | 28.07 | 0.8506 | 33.39 | 0.9432 | 31.96 | 0.8711 |
| RWKV-IR (Ours) | ×3 | 873K | 91.7G | DIV2K | 34.35 | 0.9265 | 30.20 | 0.8411 | 29.07 | 0.8044 | 28.13 | 0.8516 | 33.57 | 0.9439 | 31.95 | 0.8705 |
| SwinIR-light Liang et al. (2021) | ×3 | 886K | 87.2G | ReSyn | 34.07 | 0.9251 | 30.14 | 0.8385 | 28.96 | 0.8020 | 27.72 | 0.8427 | 32.87 | 0.9409 | 31.86 | 0.8692 |
| SRFormer-light Zhou et al. (2023) | ×3 | 861K | 105G | ReSyn | 34.15 | 0.9255 | 30.19 | 0.8386 | 28.99 | 0.8021 | 27.82 | 0.8445 | 33.07 | 0.9418 | 31.91 | 0.8695 |
| MambaIR Guo et al. (2024) | ×3 | 867K | 88.7G | ReSyn | 34.20 | 0.9260 | 30.21 | 0.8391 | 29.04 | 0.8037 | 28.04 | 0.8490 | 33.29 | 0.9430 | 32.03 | 0.8713 |
| RWKV-IR (Ours) | ×3 | 873K | 91.7G | ReSyn | 34.29 | 0.9264 | 30.14 | 0.8392 | 29.06 | 0.8038 | 28.15 | 0.8505 | 33.46 | 0.9439 | 32.05 | 0.8713 |
| SwinIR-light Liang et al. (2021) | ×4 | 897K | 49.6G | DIV2K | 31.97 | 0.8913 | 28.47 | 0.7786 | 27.48 | 0.7336 | 25.76 | 0.7753 | 30.06 | 0.9031 | 30.11 | 0.8225 |
| SRFormer-light Zhou et al. (2023) | ×4 | 873K | 62.8G | DIV2K | 32.00 | 0.8915 | 28.46 | 0.7784 | 27.48 | 0.7337 | 25.84 | 0.7780 | 30.12 | 0.9039 | 30.11 | 0.8228 |
| MambaIR Guo et al. (2024) | ×4 | 879K | 50.6G | DIV2K | 32.12 | 0.8935 | 28.51 | 0.7796 | 27.51 | 0.7339 | 25.94 | 0.7804 | 30.27 | 0.9051 | 30.22 | 0.8241 |
| RWKV-IR (Ours) | ×4 | 887K | 51.6G | DIV2K | 32.14 | 0.8942 | 28.46 | 0.7809 | 27.57 | 0.7365 | 26.07 | 0.7848 | 30.48 | 0.9074 | 30.24 | 0.8246 |
| SwinIR-light Liang et al. (2021) | ×4 | 897K | 49.6G | ReSyn | 31.93 | 0.8914 | 28.42 | 0.7778 | 27.45 | 0.7320 | 25.74 | 0.7742 | 30.04 | 0.9031 | 30.14 | 0.8222 |
| SRFormer-light Zhou et al. (2023) | ×4 | 873K | 62.8G | ReSyn | 31.97 | 0.8920 | 28.49 | 0.7782 | 27.47 | 0.7323 | 25.85 | 0.7771 | 30.16 | 0.9047 | 30.18 | 0.8230 |
| MambaIR Guo et al. (2024) | ×4 | 879K | 50.6G | ReSyn | 32.09 | 0.8938 | 28.53 | 0.7793 | 27.52 | 0.7337 | 26.02 | 0.7824 | 30.36 | 0.9070 | 30.31 | 0.8252 |
| RWKV-IR (Ours) | ×4 | 887K | 51.6G | ReSyn | 32.16 | 0.8941 | 28.45 | 0.7806 | 27.56 | 0.7351 | 26.13 | 0.7850 | 30.47 | 0.9084 | 30.34 | 0.8251 |

Table 8: Quantitative comparison on **lightweight image super-resolution** with state-of-the-art methods on 500K training iterations.

| Method | scale | #param | MACs | Set5 | | Set14 | | BSDS100 | | Urban100 | | Manga109 | |
|---|---|---|---|---|---|---|---|---|---|---|---|---|---|
| | | | | PSNR | SSIM | PSNR | SSIM | PSNR | SSIM | PSNR | SSIM | PSNR | SSIM |
| CARN Ahn et al. (2018) | ×2 | 1,592K | 222.8G | 37.76 | 0.9590 | 33.52 | 0.9166 | 32.09 | 0.8978 | 31.92 | 0.9256 | 38.36 | 0.9765 |
| IMDN Hui et al. (2019) | ×2 | 694K | 158.8G | 38.00 | 0.9605 | 33.63 | 0.9177 | 32.19 | 0.8996 | 32.17 | 0.9283 | 38.88 | 0.9774 |
| LAPAR-A Li et al. (2020) | ×2 | 548K | 171.0G | 38.01 | 0.9605 | 33.62 | 0.9183 | 32.19 | 0.8999 | 32.10 | 0.9283 | 38.67 | 0.9772 |
| SwinIR-light Liang et al. (2021) | ×2 | 878K | 195.6G | 38.14 | 0.9611 | 33.86 | 0.9206 | 32.31 | 0.9012 | 32.76 | 0.9340 | 39.12 | 0.9783 |
| SRFormer-light Zhou et al. (2023) | ×2 | 853K | 236G | 38.23 | 0.9613 | 33.94 | 0.9209 | 32.36 | 0.9019 | 32.91 | 0.9353 | 39.28 | 0.9785 |
| MambaIR Guo et al. (2024) | ×2 | 859K | 198.1G | 38.16 | 0.9610 | 34.00 | 0.9212 | 32.34 | 0.9017 | 32.92 | 0.9356 | 39.31 | 0.9779 |
| RWKV-IR (Ours) | ×2 | 859K | 198.1G | 38.22 | 0.9614 | 33.98 | 0.9210 | 32.37 | 0.9018 | 32.95 | 0.9359 | 39.34 | 0.9781 |
| CARN Ahn et al. (2018) | ×3 | 1,592K | 118.8G | 34.29 | 0.9255 | 30.29 | 0.8407 | 29.06 | 0.8034 | 28.06 | 0.8493 | 33.50 | 0.9440 |
| IMDN Hui et al. (2019) | ×3 | 703K | 71.5G | 34.36 | 0.9270 | 30.32 | 0.8417 | 29.09 | 0.8046 | 28.17 | 0.8519 | 33.61 | 0.9445 |
| LAPAR-A Li et al. (2020) | ×3 | 544K | 114.0G | 34.36 | 0.9267 | 30.34 | 0.8421 | 29.11 | 0.8054 | 28.15 | 0.8523 | 33.51 | 0.9441 |
| SwinIR-light Liang et al. (2021) | ×3 | 886K | 87.2G | 34.62 | 0.9289 | 30.54 | 0.8463 | 29.20 | 0.8082 | 28.66 | 0.8624 | 33.98 | 0.9478 |
| SRFormer-light Zhou et al. (2023) | ×3 | 861K | 105G | 34.67 | 0.9296 | 30.57 | 0.8469 | 29.26 | 0.8099 | 28.81 | 0.8655 | 34.19 | 0.9489 |
| MambaIR Guo et al. (2024) | ×3 | 867K | 88.7G | 34.72 | 0.9296 | 30.63 | 0.8475 | 29.29 | 0.8099 | 29.00 | 0.8689 | 34.39 | 0.9495 |
| RWKV-IR (Ours) | ×3 | 867K | 88.7G | 34.76 | 0.9301 | 30.59 | 0.8471 | 29.32 | 0.8096 | 29.04 | 0.8693 | 34.37 | 0.9491 |
| CARN Ahn et al. (2018) | ×4 | 1,592K | 90.9G | 32.13 | 0.8937 | 28.60 | 0.7806 | 27.58 | 0.7349 | 26.07 | 0.7837 | 30.47 | 0.9084 |
| IMDN Hui et al. (2019) | ×4 | 715K | 40.9G | 32.21 | 0.8948 | 28.58 | 0.7811 | 27.56 | 0.7353 | 26.04 | 0.7838 | 30.45 | 0.9075 |
| LAPAR-A Li et al. (2020) | ×4 | 659K | 94.0G | 32.15 | 0.8944 | 28.61 | 0.7818 | 27.61 | 0.7366 | 26.14 | 0.7871 | 30.42 | 0.9074 |
| SwinIR-light Liang et al. (2021) | ×4 | 897K | 49.6G | 32.44 | 0.8976 | 28.77 | 0.7858 | 27.69 | 0.7406 | 26.47 | 0.7980 | 30.92 | 0.9151 |
| SRFormer-light Zhou et al. (2023) | ×4 | 873K | 62.8G | 32.51 | 0.8988 | 28.82 | 0.7872 | 27.73 | 0.7422 | 26.67 | 0.8032 | 31.17 | 0.9165 |
| MambaIR Guo et al. (2024) | ×4 | 879K | 50.6G | 32.51 | 0.8993 | 28.85 | 0.7876 | 27.75 | 0.7423 | 26.75 | 0.8051 | 31.26 | 0.9175 |
| RWKV-IR (Ours) | ×4 | 879K | 50.6G | 32.53 | 0.8995 | 28.82 | 0.7875 | 27.78 | 0.7426 | 26.79 | 0.8052 | 31.28 | 0.9179 |

## C.3 Gaussian Color Image Denoising

As shown in Tab. 9, we conduct the quantitative comparison between our RWKV-IR and the SOTA methods IRCNN Zhang et al. (2017b), FFDNet Zhang et al. (2018a), DnCNN Zhang et al. (2017a), SwinIR Liang et al. (2021), Restormer Zamir et al. (2022) and MambaIR Guo et al. (2024) on long training iterations. All the models are trained on the DFWB-RGB dataset. Our method achieves competitive metrics on all four datasets.

## C.4 Grayscale Image Denoising

As shown in Tab. 10, we conduct the quantitative comparison between our RWKV-IR and the SOTA methods IRCNN Zhang et al. (2017b), FFDNet Zhang et al. (2018a), DnCNN Zhang et al. (2017a), SwinIR Liang et al. (2021) on grayscale image denoising task. All models are trained on long iterations of 500K and on DFWB-gray dataset. Our method achieves competitive metrics on all three datasets.

Table 9: Quantitative comparison on **gaussian color image denoising** with state-of-the-art methods.

| Method | BSD68 | | | Kodak24 | | | McMaster | | | Urban100 | | |
|---|---|---|---|---|---|---|---|---|---|---|---|---|
| | $\sigma$=15 | $\sigma$=25 | $\sigma$=50 | $\sigma$=15 | $\sigma$=25 | $\sigma$=50 | $\sigma$=15 | $\sigma$=25 | $\sigma$=50 | $\sigma$=15 | $\sigma$=25 | $\sigma$=50 |
| IRCNN Zhang et al. (2017b) | 33.86 | 31.16 | 27.86 | 34.69 | 32.18 | 28.93 | 34.58 | 32.18 | 28.91 | 33.78 | 31.20 | 27.70 |
| FFDNet Zhang et al. (2018a) | 33.87 | 31.21 | 27.96 | 34.63 | 32.13 | 28.98 | 34.66 | 32.35 | 29.18 | 33.83 | 31.40 | 28.05 |
| DnCNN Zhang et al. (2017a) | 33.90 | 31.24 | 27.95 | 34.60 | 32.14 | 28.95 | 33.45 | 31.52 | 28.62 | 32.98 | 30.81 | 27.59 |
| DRUNet Zhang et al. (2021b) | 34.30 | 31.69 | 28.51 | 35.31 | 32.89 | 29.86 | 35.40 | 33.14 | 30.08 | 34.81 | 32.60 | 29.61 |
| SwinIR Liang et al. (2021) | 34.42 | 31.78 | 28.56 | 35.34 | 32.89 | 29.79 | 35.61 | 33.20 | 30.22 | 35.13 | 32.90 | 29.82 |
| Restormer Zamir et al. (2022) | 34.40 | 31.79 | 28.60 | 35.47 | 33.04 | 30.01 | 35.61 | 33.34 | 30.30 | 35.13 | 32.96 | 30.02 |
| MambaIR Guo et al. (2024) | 34.44 | 31.82 | 28.64 | 35.35 | 32.92 | 29.87 | 35.63 | 33.36 | 30.32 | 35.17 | 32.99 | 30.06 |
| RWKV-IR (Ours) | 34.43 | 31.79 | 28.62 | 35.37 | 32.98 | 29.92 | 35.62 | 33.35 | 30.33 | 35.19 | 33.02 | 30.10 |

Table 10: Quantitative comparison on **grayscale image denoising** with state-of-the-art methods.

| Method | Set12 | | | BSD68 | | | Urban100 | | |
|---|---|---|---|---|---|---|---|---|---|
| | $\sigma$=15 | $\sigma$=25 | $\sigma$=50 | $\sigma$=15 | $\sigma$=25 | $\sigma$=50 | $\sigma$=15 | $\sigma$=25 | $\sigma$=50 |
| IRCNN Zhang et al. (2017b) | 32.76 | 30.37 | 27.12 | 31.63 | 29.15 | 26.19 | 32.46 | 29.80 | 26.22 |
| FFDNet Zhang et al. (2018a) | 32.75 | 30.43 | 27.32 | 31.63 | 29.19 | 26.29 | 32.40 | 29.90 | 26.50 |
| DnCNN Zhang et al. (2017a) | 33.86 | 30.44 | 27.18 | 31.73 | 29.23 | 26.23 | 32.64 | 29.95 | 26.26 |
| DRUNet Zhang et al. (2021b) | 33.25 | 30.94 | 27.90 | 31.91 | 29.48 | 26.59 | 34.44 | 31.11 | 27.96 |
| SwinIR Liang et al. (2021) | 33.36 | 31.01 | 27.91 | 31.97 | 29.50 | 26.58 | 33.70 | 31.30 | 27.98 |
| MambaIR Guo et al. (2024) | 34.44 | 31.82 | 28.64 | 35.35 | 32.92 | 29.87 | 35.63 | 33.36 | 30.32 |
| RWKV-IR (Ours) | 34.46 | 31.85 | 28.66 | 35.33 | 32.90 | 29.84 | 35.64 | 33.35 | 30.34 |

Table 11: Quantitative comparison on **JPEG compression artifact reduction** with state-of-the-art methods. We show scores of average PSNR/SSIM/PSNR-B.

| Method | Classic5 | | | | LIVE1 | | | |
|---|---|---|---|---|---|---|---|---|
| | $q$=10 | $q$=20 | $q$=30 | $q$=40 | $q$=10 | $q$=20 | $q$=30 | $q$=40 |
| ARCNN Zhang et al. (2018c) | 29.03/0.7929/28.76 | 31.15/0.8517/30.59 | 32.51/0.8806/31.98 | 33.32/0.8953/32.79 | 28.96/0.8076/28.77 | 31.29/0.8733/30.79 | 32.67/0.9043/32.22 | 33.63/0.9198/33.14 |
| DnCNN-3 Zhang et al. (2017a) | 29.40/0.8026/29.13 | 31.63/0.8610/31.19 | 31.63/0.8610/31.19 | 33.77/0.9003/33.20 | 29.19/0.8123/28.90 | 31.59/0.8802/31.07 | 32.98/0.9090/32.34 | 33.96/0.9247/33.28 |
| DRUNet Zhang et al. (2021b) | 30.16/0.8234/29.81 | 32.39/0.8734/31.80 | 33.59/0.8949/32.82 | 34.41/0.9075/33.51 | 29.79/0.8278/29.48 | 32.17/0.8899/31.69 | 33.59/0.9166/32.99 | 34.58/0.9312/33.93 |
| SwinIR Liang et al. (2021) | 30.27/0.8249/29.95 | 32.52/0.8748/31.99 | 33.73/0.8961/33.03 | 34.52/0.9082/33.66 | 29.86/0.8287/29.50 | 32.25/0.8909/31.70 | 33.69/0.9174/33.01 | 34.67/0.9317/33.88 |
| RWKV-IR (Ours) | 30.35/0.8261/30.04 | 32.63/0.8760/32.05 | 33.81/0.8972/33.12 | 34.61/0.9091/33.71 | 29.94/0.8296/29.62 | 32.34/0.8915/31.81 | 33.78/0.9185/33.12 | 34.78/0.9323/33.95 |

## C.5 JPEG COMPRESSION ARTIFACT REDUCTION

Tab. 11 shows the comparison of RWKV-IR with state-of-the-art JPEG compression artifact reduction methods: ARCNN Zhang et al. (2018c), DnCNN-3 Zhang et al. (2017a), DRUNet Zhang et al. (2021b) and SwinIR Liang et al. (2021). Following Zhang et al. (2021b); Liang et al. (2021), we test different methods on two benchmark datasets (Classic5 and LIVE1) for JPEG quality factors 10, 20, 30 and 40. It can be seen that the proposed RWKV-IR has average PSNR gains of at least 0.07dB and 0.08dB on two testing datasets for different quality factors.

