# OpenReview forum: "Real&Synthetic Dataset and the Linear Attention in Image Restoration"
_ICLR.cc/2025/Conference — ICLR 2025 Conference Withdrawn Submission_

### Official Review · Reviewer_zzf6 · 2024-10-29

**Soundness:** 2
**Presentation:** 3
**Contribution:** 2
**Rating:** 5
**Confidence:** 5

**Summary:**

This study proposes a large-scale IR dataset ReSyn for image restoration (IR) tasks, which includes real and AIGC synthesized images, to address the lack of a unified training benchmark and image complexity distribution bias in existing methods. Introducing an image filtering method based on image complexity, balancing the distribution of image complexity, and constructing a unified training standard. In addition, the RWKV-IR model is proposed, which combines linear complexity RWKV and Transformer-based image restoration structure to achieve linear attention for global and local perception through deep convolution displacement and bidirectional attention mechanism.

**Strengths:**

1. This manuscript proposes the ReSyn dataset to address the issue of complexity distribution bias between training and testing datasets in image restoration tasks.
2. The RWKV-IR model is proposed, which integrates linear complexity RWKV and Transformer architecture to improve image restoration performance.
3. The experiment shows that the RWKV-IR model is competitive in image restoration and can effectively handle global and local features.

**Weaknesses:**

1. The manuscript mentions a strong Pearson correlation between the GLCM complexity measure and BPP (Bits Per Pixel), but the significance of this correlation in the context of your image complexity measure and its predictive power for PSNR (Peak Signal-to-Noise Ratio) metrics is not explicitly discussed. Could you elaborate on how this correlation supports the effectiveness of your image complexity measure as a predictor for PSNR?
2. This manuscript conducts image complexity analysis based on Gray Level Co occurrence Matrix (GLCM) and points out that it is closely related to the sensitivity of the human eye to texture. However, some statistical measures of GLCM, such as Entropy, Energy and Dissimilarity, may exhibit inconsistent performance on different types of images. Have you conducted sufficient testing on different types of images to ensure the universal applicability of GLCM complexity measurement?
3. The author mentions that 'GLCM complexity measure has a strong Pearson correlation compared to BPP', but does not explicitly state the actual significance of this correlation. Please further explain how this correlation supports stronger prediction of your image complexity measure as a PSNR metric.
4. The author introduces RWKV and improved its core components to enhance its effectiveness in image restoration tasks. However, its role is not yet clear, and it is recommended to add strategies such as feature visualization to further verify the role of RWKV in restoration tasks.
5. The experiments are not sufficient. I am curious about the reaction if we replace the Spatial Mixer or Channel Mixer in RWKV with other components? For example, replacing Spatial Mixer with Multi head Self Attention (MHSA) or SSM2D.

**Questions:**

See the above Weaknesses part.

---

### Official Review · Reviewer_fBGc · 2024-11-02

**Soundness:** 2
**Presentation:** 1
**Contribution:** 2
**Rating:** 3
**Confidence:** 5

**Summary:**

This work introduces a large-scale dataset ReSyn with balanced image complexity distribution between training and test datasets. To evaluate image complexity, the authors introduces a Gray-Level Co-occurrence Matrix based metric to filter images for balanced complexity. Additionally, the authors present a transformer model RWKV-IR and introduces a unified IR training benchmark to standardize model evaluation.

**Strengths:**

1. This work constructs a large-scale dataset called ReSyn, that integrates both real and synthetic images. It considers the AIGC images as an essential part of the dataset.
2. This work introduces a unified benchmark to assess IR models' convergence and restoration capabilities, evaluating them on both the ReSyn and other commonly used datasets.

**Weaknesses:**

1. The experimental analysis was predominantly concentrated on the ablation studies of the proposed model, which resulted in an oversight of a detailed examination into the newly introduced metrics, the challenges posed by imbalanced complexity, and an in-depth analysis of the mixed dataset's properties.
2. The paper does not address the potential benefits or drawbacks of using AIGC images in the ReSyn dataset, which is a significant omission since understanding their impact could reveal important insights into model performance and generalization capabilities across different image types.
3. How do the complexities of the proposed model and the compared models differ when applied to various tasks such as super-resolution, image denoising, and JPEG artifact reduction?
4. While the proposed model shows some feasible results, it still lags behind current SOTA models in most tasks according to the tables.
5. In the related work, there is a lack of sufficient discussion on the distinctions between existing works and this work. Without a detailed comparative analysis, it becomes challenging to understand the advantages of this work over the existing ones.
6. Figure 4 has an issue with inconsistent color usage for blocks, where the same block is depicted in different colors and distinct blocks share the same color. Additionally, Fig.4(a) abbreviates the name of Fig.4(b) without providing any indication.
7. Many writing problems such as 'global and receptive fields', 'replace the the original', 'stored' in Fig.3, and the meaning of the markers in Fig.3 is not explained.

**Questions:**

Please refer to the above Weaknesses.

---

### Official Review · Reviewer_aQVz · 2024-11-03

**Soundness:** 3
**Presentation:** 3
**Contribution:** 2
**Rating:** 5
**Confidence:** 5

**Summary:**

This paper analyzes image complexity based on the Gray Level Co-occurrence Matrix (GLCM) and combines AIGC-generated images with real images to create a large-scale infrared dataset, ReSyn, which is more suitable for image restoration tasks. Additionally, the RWKV-ir model is proposed, using DC-shift to replace the original Q-shift for better adaptation to low-level vision tasks. A Cross-Bi-WKV module is also introduced to address the attention imbalance of the model in horizontal and vertical directions.

**Strengths:**

1. This paper is well-written, with clear and accessible illustrations.

2. The design of the deep convolutional shift and Cross-Bi-WKV is simple yet effective.

**Weaknesses:**

1. AIGC images are generated images, which generally have shortcomings in local details. Additionally, there is a lack of experimental validation regarding the specific role of such images.

2. The dc-shift principle used in the paper is similar to auto-correlation but is relatively simplistic, lacking experimental and generalization tests.

**Questions:**

1. What is the proportion of AIGC images in the dataset? Since AIGC images are generated, do they contribute low-resolution images as part of the dataset?

2. How are the convolution kernels for the dc-shift deep convolutional shift selected? Moreover, in Figure 5, the dc-shift shows the four pixels in the diagonal direction as light green but does not explain the reason—does this indicate weaker correlation or has weight scaling been applied?

3. Is there an error in Equation 3? The standard WKV calculation formula does not have a denominator; such an obvious error should not occur. If it is not an error, please explain the formula. Additionally, a similar normalization operation is performed by dividing by T—why is the position encoding information represented by u not divided by T?

4. Is it based on RWKV-v4 or v6? Has it been tested on higher resolution image restoration and enhancement tasks? RWKV is known for its efficiency; can you provide corresponding runtime and comparison?

5. Vision RWKV, VIT, and others have demonstrated that even simple MLPs can achieve token mixing and perform well. Why are they considered unsuitable for image restoration? Please provide reasons.

---

### Note · Authors · 2024-11-15

I have read and agree with the venue's withdrawal policy on behalf of myself and my co-authors.

---

> ### Author Response · Authors · 2024-11-15
>
> Thanks to the reviewers' valuable feedback, we have decided to withdraw our paper to make some revisions.